# Composite Graph Neural Networks for Molecular Property Prediction

**DOI:** 10.3390/ijms25126583

**Published:** 2024-06-14

**Authors:** Pietro Bongini, Niccolò Pancino, Asma Bendjeddou, Franco Scarselli, Marco Maggini, Monica Bianchini

**Affiliations:** Department of Information Engineering and Mathematics, University of Siena, 53100 Siena, Italy; pietro.bongini@unisi.it (P.B.); niccolo.pancino@unisi.it (N.P.); biofamily@live.it (A.B.); franco@diism.unisi.it (F.S.); marco.maggini@unisi.it (M.M.)

**Keywords:** artificial intelligence, deep learning, graph neural networks, molecular property prediction, composite graph neural networks, open graph benchmark, molecular graphs

## Abstract

Graph Neural Networks have proven to be very valuable models for the solution of a wide variety of problems on molecular graphs, as well as in many other research fields involving graph-structured data. Molecules are heterogeneous graphs composed of atoms of different species. Composite graph neural networks process heterogeneous graphs with multiple-state-updating networks, each one dedicated to a particular node type. This approach allows for the extraction of information from s graph more efficiently than standard graph neural networks that distinguish node types through a one-hot encoded type of vector. We carried out extensive experimentation on eight molecular graph datasets and on a large number of both classification and regression tasks. The results we obtained clearly show that composite graph neural networks are far more efficient in this setting than standard graph neural networks.

## 1. Introduction

Graphs are a ubiquitous and very important form of data representation, providing information on data entities— represented by nodes—and their relationships, represented by edges. Graph neural networks (GNNs) have become a gold standard for solving problems defined on graphs. These powerful models, first introduced in 2008 [1], are designed to process graphs, replicating their structure in the model architecture. Moreover, thanks to their mathematical formulation, GNNs ensure minimal loss of structural information and have been shown to be universal approximators on graphs. In order to measure the computational capabilities of GNNs, the unfolding tree method and the Weisfeiler–Lehman [2] test are used. These methods, which were demonstrated to be equivalent, also allow for the classification of different models according to the expressive power they can reach [3]. It is always recommended to use models that are at least as capable of distinguishing isomorphic graphs as the Weisfeiler–Lehman test of the first order (WL–1 class). GNNs can be used for both the regression and classification of properties defined on the nodes (or a subset thereof) in a dataset composed of one or more graphs, as well as the property regression and classification of edges (or a subset thereof) and the property regression and classification of graphs. Additionally, they can efficiently solve link prediction and ablation problems, as well as graph generation tasks. In recent years, GNNs have experienced a consistent spike in popularity and have been applied to a huge variety of different problems, covering almost every field of science and technology [4,5,6]. This great versatility is due to the large variety of models that have been developed, belonging to two large families—recurrent GNNs and graph convolution networks (GCNs)—and it is also thanks to the well-founded mathematical properties of GNNs [7]. Convolutional models diffuse and pool information over a graph by replicating a convolution operator over the neighborhoods. This can be carried out in both the spatial [8] and spectral domains [9,10]. The relevant models of this family also include graph attention networks (GATs) [11], which exploit a graph attention mechanism over graph nodes, and GraphSAGE [12], which uses various aggregation mechanisms, including long short–term memories (LSTMs), to make the information flow between node neighbourhoods. In contrast, recurrent GNNs make the information flow through the graph structure using a message-passing mechanism. Along with the original model [1], they include message-passing neural networks (MPNNs) [13] and graph isomorphism networks (GINs) [3]. Moreover, recurrent GNNs can also be stacked into a more complex architecture known as Llyered GNN (LGNN), which iteratively refines the outputs of the first GNN in the stack, using the upper GNNs to improve the understanding of the problem and, consequently, the proposed solution. GNNs of every type have been employed in a wide variety of different applications, from social network analysis [14] and spam node detection on the web to weather forecasting [15], power network analysis [16], and telecommunication network optimization [17]. In the biological domain, GNNs and GCNs have been applied to drug discovery [18], the prediction of compound mutagenicity [19], anti–HIV activity [20,21], and protein–protein interactions [22], just to name a few tasks. In particular, our method [23], which is specifically well-suited to biological applications [24], has been successfully applied to the following: molecular graph generation [25], integrating the generation method to a full pipeline of GNN-based filters for candidate selection in the drug discovery domain [26]; drug side-effect prediction on a heterogeneous graph integrating many different sources of information [27], on molecular graphs only [28], and on a combination of these two setups; the identification of protein–protein interactions [22]; link prediction for suggesting possible matches in a caregiver support network [29].

In this paper, we will explore the capabilities of composite graph neural networks (CGNNs) in molecular analysis tasks. CGNNs are a variant of the original recurrent GNN model [1], in which a state-updating network is defined for every type of node in the graph. While the predictive capabilities of GNNs have been widely studied [7], the predictive capabilities of CGNNs have not been thoroughly analyzed so far. As a consequence, we propose a comparison between the standard and composite versions of the original recurrent GNN model [1]. We use some very well-known and solid benchmark datasets that are part of the open graph benchmark (OGB) [30]. Each dataset is composed of molecular graphs. The tasks are also defined by the OGB and are a mix of the classification of molecules based on their activities or categories and the regression of molecular properties. In some cases, multiple tasks are required to be carried out in parallel. The main contributions of the paper are as follows: a comprehensive method based on GNNkeras [23] to train and evaluate CGNNs and GNNs on graph-focused regression or classification tasks on any molecular graph dataset; an extensive comparison between CGNNs and GNNs on benchmark datasets taken from the OGB.

The rest of this paper is organized as follows. Section 4.1 describes the datasets and associated learning tasks on which we trained and tested our model. Section 4.2 describes the GNN and CGNN models and explains the basics of the message-passing mechanism behind recurrent GNNs. Section 4.3 explains the experimental methodology, model hyperparameters, and technical features. Section 2 presents the experimental results and discusses their significance. Finally, Section 3 draws the conclusions of the research work and explains how it can be useful for future investigation.

## 2. Results

The evaluation was carried out using the "Evaluator" provided with the OGB package; the metrics are determined by the OGB itself and vary according to the dataset. This occurs both in the grid-search phase, where multiple configurations of the same model are validated, and in the test phase, where the best configurations of the two models are compared. The best configurations obtained on every dataset are summarized in Table 1, reporting the hyperparameters of the best models.

In particular, the evaluation metrics used by the OGB Evaluator are area under receiver operator characteristic curve (AUROC) and average precision (AP) for binary classification problems, and it uses the root mean squared error (RMSE) for regression problems. These metrics are the standard metrics provided by the OGB for the evaluation of models on their datasets. In particular, AUROC is used for its robustness to dataset imbalancement. The AUROC metric measures the area under the ROC curve, a function that associates a true positive rate (TPR) value with each false positive rate (FPR) value, which is possible given the predictions of the model, and we varied the classification threshold over the [0, 1] interval. The expected curve for a random classifier is the line representing the function TPR = FPR. In order to do better, the curve must trend towards a function that grows rapidly and then stabilizes just below the TPR = 1 line. A perfect classifier would have a curve corresponding only to TPR = 1. The area under such a curve, therefore, gives a score (between 0 and 1) that accounts for how accurate the classifier is. For a random classifier, AUROC = 0.5 is expected. A classifier should, therefore, perform better than AUROC = 0.5 in order for it to be acceptable (better than random guessing). The AP metric measures the average value of precision across a set of parallel binary classification tasks instead. The precision is measured as the ratio of true positives over the sum of true positives and false positives. Finally, the RMSE is the (positive) root of the MSE between the predicted values and target values. The results are reported in Table 2, showing the AUROC, RMSE, or AP of the best model on every dataset. The metric to be used depends on the dataset and is defined by the OGB benchmark itself to best represent the model’s predictive capacity depending on the type of task and on data unbalance.

The results clearly show the superiority of the CGNN on a large majority of molecular datasets, with BBBP representing the only exception by a narrow 0.003 AUROC gap. This confirms that CGNNs are capable of exploiting the heterogeneous nature of molecular graphs and can, therefore, better classify molecules according to their activity and predict their properties with a smaller error. Moreover, this is true for both single-task problems, in which only one property is predicted and evaluated, and for multi-task problems, where a multi-class-multi-label approach is required to predict a set of the parallel, relevant properties of the graphs in the dataset. The better results of CGNNs were obtained thanks to the specialization of the state-updating network on more precise subtasks. Each MLP specializes in a single node type, allowing for a more accurate estimation of the following node state given all the input quantities (current node state, the current state of the neighbors, and the label of the node, its neighbors, and the relative edges). This specialization of the MLPs allows for a better local understanding of the graph structural properties and node/edge labels, which translates to a better classification/regression performance by the output network (which, instead, is not specialized on node types). This benefit comes with a computational overhead, with the model’s complexity (in terms of time) increasing due to the fact that more MLPs are utilized. The complexity in terms of memory increases because more neural networks need to be kept in memory, yet the state-updating MLPs typically require fewer parameters because their task is more specialized. This paper proves the properties of CGNNs for graph-focused molecular problems, yet such properties are expected to also hold on node-focused problems, in which the output is defined on a subset of nodes $N_{out}\subseteq N$, and on edge-focused problems, where the output is defined on a subset of edges $E_{out}\subseteq E$. These two categories of problems can be seamlessly solved by using the same GNN and CGNN models and by changing only the output function, $g_{w}$, which is still calculated with a single (non-specialized) output network. Having a set of output networks, with each one specialized in approximating the output function, $g_{w}$, on types of nodes or edges, is expected to bring an advantage proportional to the one introduced by the different state-updating networks. Extending these results to node-focused and edge-focused problems will be a matter of future research.

As an additional measure of the significance of the results obtained, we propose a comparison between our best models and the current state-of-the-art (SotA) methods on each specific task. In principle, domain-specific applications can be tuned to the problem much more than a general model, thus obtaining much better results. However, our CGNN model can be easily integrated and hybridized with other deep learning solutions, refining the architecture for a specific task and improving its performance. For instance, even just applying the Layered graph neural network paradigm to our CGNNs would very likely improve the performance level. We have collected the SotA methods for the tasks for analysis in Table 3, and we compared them to our CGNN models.

As shown in Table 3, different datasets require different model characteristics to extract the relevant information from the molecular graphs. GINs [3] and GCNs [8] expectedly represent the best methods in most tasks. On HIV, instead, a domain-specific adaptation of Graphormer [31] outperforms all the other methods. CGNNs outperform previous SotA methods on three datasets out of eight. In particular, they outperform GCNs on BACE, ClinTox, and Sider. On the latter dataset, standard GNNs are perfectly on par with GCNs, while CGNNs outperform both. These experimental results are in line with the expectations because CGNNs share the same theoretical properties of standard GNNs, including WL-1 isomorphism recognition and universal approximation on graphs, while they also exploit the heterogeneous nature of molecular graphs, thanks to the specialization of state-updating MLPs. This makes CGNNs capable of outperforming WL-0 GCNs in most cases, while the latter is still better for some classes of graph-focused tasks, thanks to graph pooling operations that are not available for recurrent GNNS and CGNNs.

## 3. Discussion

This work proposes a comparison between the standard recurrent GNN model, operating on homogeneous versions of molecular graphs, and the CGNN model, which processes heterogeneous graphs and maps the atoms of different species to different node types. Both models are defined and implemented using the GNNkeras framework [23]. Eight molecular graph datasets from the OGB [30] were used in the experimentation; we compared the models for the regression of molecular properties, the classification of molecules according to their activity, and in a parallel multi-task approach. The experimental results clearly show that CGNNs significantly outperform standard GNNs on all datasets but one (where they perform similarly), confirming that the capability of processing heterogeneous graphs is key for molecular property prediction. This holds true for both regression and classification tasks and also in multi-class, multi-label classification scenarios. Moreover, a comparison with previous SotA methods on each dataset demonstrated that CGNNs outperform them in three cases out of eight, showing that the CGNN paradigm combined with the approximation capabilities of recurrent GNNs can boost current learning methodologies on molecular graphs in a variety of scenarios. The key conclusion of this work is that using CGNNs can boost the capabilities of GNNs with regard to many biological problems. This comes with a computational overhead that is absolutely manageable due to the fact that GNNs are very lightweight in comparison to many other deep learning models, and thanks to the specialization of the state-updating networks contained in CGNNs. This is, of course, a partial result, as the same capabilities need to be proved in the future in many other application fields where GNNs could be potentially useful, e.g., the analysis of social networks, knowledge graphs, and power networks. Yet, given the importance of the biological problem and the volume of application of GNNs to such complex environments, this represents a key step towards a more generalized methodology for tackling problems on heterogeneous graphs. In the biological domain, for instance, a direction to be investigated in the near future is that of protein structures, which can be represented by graphs composed of aminoacids. In this scope, it will also be possible to prove the same properties of CGNNs on node-focused problems, for instance, by applying CGNNs to protein–protein interface identification and edge-focused problems, such as polypharmacy effect prediction, in which drug nodes can be linked according to the probability they have of interacting with each other. Moreover, as graph generation algorithms are becoming more efficient and specialized, investigating the impact of heterogeneous graph processing would also be very interesting in this application field.

## 4. Materials and Methods

In this section, we describe every aspect of the methodology of our study. In particular, we will introduce the datasets, describe the models developed in this work, and discuss the experimental setup.

### 4.1. Datasets

We evaluated our method on eight molecular graph benchmarks from the open graph benchmark (OGB) collection [30]. The OGB is a widely used and high-quality repository for graph-based model development and evaluation. The OGB proposes datasets of different sizes for every graph-based task. Both classification and regression tasks—focused on nodes, edges, and whole graphs—are provided. Multi-class classification, multi-task regression, and link prediction problems are available as well. Moreover, though downloaded and evaluated through the OGB platform, all datasets come from the MoleculeNet project [32]. Both classification and regression tasks were addressed, sometimes with multiple tasks being carried out in parallel on the same dataset. Table 4 lists all the datasets, together with their main characteristics, used in our experimentation and the number and type of tasks. In particular, for HIV, the task is to predict if a molecule has anti-HIV activity. This task is unbalanced because most molecules belong to the negative class (no activity). FreeSolv consists of a graph-based regression of the free energy of solvation of the molecule represented by the graph. In Tox–21, we aimed to identify 12 (nonmutually exclusive) categories of toxicity for the compounds in the dataset. In BACE, the objective is to identify which of the molecules are inhibitors of human β–secretase 1. On the BBBP dataset, we aimed to identify which molecules can penetrate the blood–brain barrier, therefore representing possible drugs for the brain. For ClinTox, the target is to classify molecules based on two nonmutually exclusive macro-categories of toxicity. Regarding MUV, we aimed to classify compounds based on 17 different virtual screening compliance classes. Again, the classes are not mutually exclusive and, therefore, require (binary) multi-class multi-label classification. Finally, concerning the Sider dataset, we aimed to produce a (binary) multi-class multi-label classification of the drugs contained in the dataset based on their adverse reactions, grouped into 27 system–organ nonmutually exclusive classes.

We downloaded the datasets through the OGB Python package (https://pypi.org/project/ogb/) (accessed on 25 April 2024) and used the standard dataset splits defined by the OGB [30]. As a consequence, the training–validation–test split is the standard one provided by the benchmark, and it is the same throughout all the experimentation. Moreover, the percentages of the training set examples, validation set examples, and test set examples are 80%, 10%, and 10%, respectively, for every dataset. The validation set is used for an early stopping module, which prevents overfitting and stops the training procedure when validation loss starts to increase. The early stopping module has a patience of 10, and it is applied after every training epoch. We preprocessed OGB graphs in order to send them as input to GNNs and CGNNs, transforming them into “GraphObjects” and “CompositeGraphObjects”, respectively. These two latter data types are defined by GNNkeras and allow for the optimal processing of the graphs using the GNN and CGNN models [23]. The CGNNs were given heterogeneous molecular graphs as input, with each atom species mapped to a dedicated node type. Standard GNNs, which can only process homogeneous graphs, were, instead, given a homogeneous version of the molecular graphs as input, with one-hot encoded vectors representing the atom species as node labels. Moreover, both node types for CGNNs and the corresponding one-hot encodings for GNNs were grouped according to atom groupings on the periodic table; this step was necessary to avoid using too many node types and excessively long one-hot vectors. In particular, many atom species appear only a handful of times throughout the dataset, making them impossible to learn. Once grouped, they amount to reasonable quantities instead. Since the grouping follows the classes of equivalence defined by the periodic table of elements, the atom species mapped to the same type will have similar chemical behavior. The most common atom species have their own group, while uncommon atom species with similar characteristics were put together, obtaining a total of eight element groups: 1—Metals, 2—Metalloids, 3—Halogens, 4—Carbon, 5—Nitrogen, 6—Oxygen, 7—Phosphorus, and 8—Sulfur.

### 4.2. Model

Due to its expressive power, we used the original recurrent GNN model [1] for our experimentation. In particular, we exploited its recent Tensorflow2–Keras implementation, known as GNNkeras [23]. GNNs process structural information by calculating a state over every node. The node state should acquire all the relevant information on the node itself, the local graph structure, and the features of the edges and neighbor nodes. In an iterative process called “message passing,” each node sends its state to all of its neighbors and receives the states from all of them. In order to calculate the state value at the following time instant (iteration), all the incoming messages are aggregated, encapsulating information on the edges on which they traveled. The node state at the previous iteration and the aggregated messages are fed as input to a state-updating network, which produces the new node state in output. The state-updating network is a neural network (usually an MLP). This network is copied on every graph node, acting as a sort of building block to make up the architecture of the GNN (which retraces the input graph structure). All of the copies of the state-updating network share the same weights, limiting the number of parameters of the model and preventing the vanishing gradient problem. The message-passing process is sketched in Figure 1. After a fixed number of message-passing iterations, an output function is calculated. In order to do so, a dedicated output network was replicated over the nodes or the edges, similar to what was carried out using the state-updating network. The output network is another MLP, and all of its copies share the same weights. If the problem is node-focused, the output network is replicated over every node (or a relevant subset of graph nodes). Each copy takes only the final state (after the last message-passing iteration) of the corresponding node. If the problem is graph-focused, the output is calculated in the same way, and it is then aggregated on the whole graph by summing or averaging the single node outputs. If the problem is edge-focused instead, the output network is replicated over every edge (or a relevant subset of graph edges). In this case, its input is composed of the state of both nodes connected by the corresponding edge concatenated to the edge feature vector. The network composed of all the state-updating MLP copies and output MLP copies is called the encoding network; this replicates the structure of the input graph in its architecture. This network also unfolds in time, replicating each state-updating MLP copy once for every message-passing iteration. The structure we obtain from this latter step is called the unfolded encoding network, and it is the architecture on which the backpropagation through the structure algorithm is applied [1].

GNNkeras defines a standard GNN model, corresponding to the original formulation described above, and a composite GNN (CGNN) model, which is an adaptation of the original model for heterogeneous graphs. Heterogeneous graphs are graphs that have nodes that belong to different types according to their properties or different natures. In chemistry and biology, molecular graphs are heterogeneous by nature, being composed of atoms of different species. These species of atoms are mapped to one-hot feature vectors for standard GNNs and to different node types for CGNNs (see Section 4.1 for the description of node types). CGNNs process each type of node with a dedicated state-updating network, therefore learning a specialized version of the state-updating function for each different node type. In the following, we have reported the equations that define both models. For a more detailed description of the models, please refer to the GNNkeras paper [23].

When given a graph, G=(N,E), where *N* is the set of nodes and E={(n,m):n,m∈V} represents the set of edges, we can define a neighborhood function Ne(n)=m∈N:(n,m)∈E that maps every node to the set of its neighbors. Nodes can be associated with labels $l_{n}\;\forall n\in N$, describing their properties. Edges can be associated with labels $e_{m,n}\;\forall \left(m,n\right)\in E$, describing the corresponding relationships.

We can define a GNN as a model that approximates an output function, $g_{w}$, that can be defined on the nodes or a subset of them, $N_{out}\subseteq N$, on the edges or a subset of them, $E_{out}\subseteq E$, or on the whole graph, *G*. In our case, $g_{w}$ will always be defined as a property of the whole graph, *G*. In order to calculate this approximation, the GNN will process the graph structure and the labels (features) of the nodes and edges. The GNN will associate a state, $x_{n}$, to each node, n∈N, which will be iteratively updated by a learnable state-updating function, $f_{w}$. The state is a vector of dimension $d_{x}$, which is set as a hyperparameter of the GNN and initialized by sampling from a random distribution, usually centered on the origin of $R^{d_{x}}$. The states are updated for *K* iterations, where *K* is a model hyperparameter. Given the randomly sampled initial states $x_{n}^{0},\;\forall n\in N$, the state of any node, *n*, at iteration *t* can be calculated using the state-updating function $f_{w}$, as in Equation (Equation 1):(1)$$x_{n}^{t}=f_{w}\left(x_{n}^{t-1},\;l_{n},\sum_{m\in Ne(n)}\left(x_{m}^{t-1},\;l_{m},\;e_{m,n}\right)\right)$$
When the maximum number of iterations, *K*, is reached, the final versions of the node states, $x_{n}^{K},\;\forall n\in N$, are fed as input into the output network, which approximates the output function, $g_{w}$. The formulation of the output function depends on the type of problem. In this paper, since only graph-focused problems are addressed, we only report the graph-focused version of $g_{w}$, as described in Equation (Equation 2):(2)$$y_{G}=\frac{1}{|N_{out}|}\sum_{n\in N_{out}}g_{w}\left(x_{n}^{K},\;l_{n}\right)$$

As can be seen in Equation (Equation 2), the output is calculated on each node, $n\in N_{out}$, for which the output is defined. The contributions of all the nodes are then averaged throughout the whole graph, obtaining a global graph output, $y_{G}$. This allows for an analysis of the graph structure from multiple different local points of view. The local results are then merged, obtaining a global understanding of the graph comprising both the graph structure and the features of nodes and edges. In the standard GNN model, an MLP called the state-updating network is dedicated to the calculation of $f_{w}$, whereas another MLP, called the output network, is dedicated to the calculation of $g_{w}$. In composite graph neural networks (CGNNs), there are multiple state-updating networks, with each calculating a dedicated version, $f_{w,i}$, of the state-updating function for a node of type *i*. In order to map each node to its type, a function, T(n), is defined, associating a type *i* to *n*. Remarkably, all the state-updating functions, $f_{w,i}$, have the same output dimension, corresponding to the state dimension $d_{x}$; this allows nodes of different types to exchange messages in a seamless way. The state-updating function can, therefore, be rewritten as in Equation (Equation 3):(3)$$x_{n}^{t}=f_{w,i}\left(x_{n}^{t-1},\;l_{n},\;a\sum_{m\in Ne(n)}\left(x_{m}^{t-1},\;e_{m,n}\right)\right)\;:\;i=T\left(n\right)\;.$$

Of course, we will have a number of $f_{w,i}$ and, consequently, a number of state-updating networks, which is equal to the number of node types present in the dataset. The output function and the output network are identical to the standard case instead. Since the node states are all the same size, one output network is sufficient to calculate $g_{w}$, even in the heterogeneous case [33].

### 4.3. Experiments

We carried out a comparison between the performance of CGNNs and standard GNNs on each dataset. A grid-search procedure was applied to find the best configuration of each model on every dataset; then, we compared the best configurations of the two models. The grid search was performed on a set encompassing all of the hyperparameters that have a key role in determining model performance. Both models were equipped with a state-updating MLP (with a single hidden layer) and an output MLP (with a single hidden layer). We also tried configurations with more than one hidden layer, but these never performed on par with single-layer versions, demonstrating that these were not fit for this particular problem setting. The hidden layer of both networks and the output layer of the state-updating network share the same activation function, which was chosen as a hyperparameter. The output layer of the output network is equipped with a logistic sigmoid for both binary classification and regression problems (in which the output is normalized accordingly). The hyperparameters and their values are reported in Table 5. The other hyperparameters were kept fixed throughout all the experiments. In particular, we trained every configuration with the Adam optimizer [34] for a maximum of 300 epochs, with a maximum number of GNN state-updating iterations equal to 6; we used the *sum* as the aggregation function for incoming messages from neighbor nodes. The optimizer is equipped with an early stopping module based on validation loss; this checks the model every epoch, with a patience of 10. In case of early stopping, the best model parameters (previously saved after the iteration with the best validation loss) are restored.

The same hyperparameters and values were taken into account on every dataset. Since all the combinations of hyperparameter values were validated in all the grid-search experiments, each of the latter consisted of training and validating a total of 960 GNN model configurations and 960 CGNN model configurations. Each model underwent this experimentation, thanks to a Python script that automatically initialized, trained, and tested each of the 960 configurations, using GNNkeras [23] to build and run the GNN and CGNN models. In summary, for the grid-search experiments carried out on the eight datasets, we trained and tested 8 × (960 + 960) = 15,360 models. Nevertheless, the computational burden was not too heavy. GNNs are very lightweight [1] and usually require a small amount of parameters to generalize a task [23]. CGNNs follow the same computation scheme as regular GNNs but with a larger number of state-updating MLPs (one for every node type). These state-updating MLPs, however, require fewer parameters to generalize their more specialized tasks, and most importantly, they are applied only to the subset of nodes of their type. As a worst-case scenario, we can consider the computational complexity of a CGNN to be roughly twice the computational complexity of the corresponding standard GNN, while memory complexity is multiplied by the number of node types for what concerns the network parameters, and this is the same for what concerns the input data. As a consequence, both models are very easy to apply to most graph datasets. For instance, our entire experimentation (as described above) was carried out on a single Nvidia 3080 GPU, and it took a cumulative time of 19 d, 3 h, and 43 m to carry out all the 15,360 training + evaluation procedures; the approximate average was 1.8 min per experiment. The computation requirements and also the difference in performance between the two models vary according to the dataset. On average, the CGNN models took approximately 2.3 min per experiment to be trained and tested, while the standard GNN models took approximately 1.3 min per model.

## Figures and Tables

**Figure 1 ijms-25-06583-f001:**
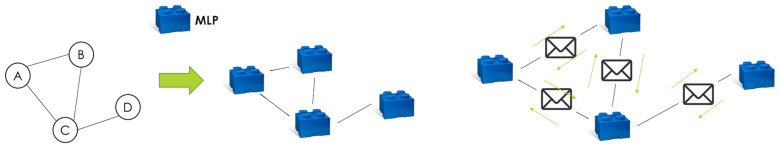
The graph neural network model places a copy of the state-updating network over every node of each input graph. After a fixed number of “message-passing” iterations, in which the state of every node is updated based on its previous state, the previous states of its neighbors and the node and edge labels as output are calculated based on the final state. The state-updating network copies share their weights and can be seen as twin building blocks that compose the adaptive architecture of the GNN, replicating the structure of the input graph.

**Table 1 ijms-25-06583-t001:** Hyperparameter values of the best GNN and CGNN configuration on every dataset. Hyperparameters are defined as the following: Initial learning rate (ILR), hidden units of state updating network (HS), hidden units of output network (HO), state dimension (SD), and activation function (AF). See Table 5 for the hyperparameter value combinations used in all the experiments.

Dataset	Best GNN					Best CGNN				
	**ILR**	**HS**	**HO**	**SD**	**AF**	**ILR**	**HS**	**HO**	**SD**	**AF**
HIV	$10^{-2}$	30	70	10	tanh	$10^{-3}$	50	20	5	tanh
FreeSolv	$10^{-2}$	20	40	30	tanh	$10^{-2}$	10	100	30	selu
Tox-21	$10^{-2}$	50	20	15	relu	$10^{-2}$	50	40	30	relu
BACE	$10^{-3}$	50	20	10	relu	$10^{-2}$	30	20	30	selu
BBBP	$10^{-2}$	10	40	10	tanh	$10^{-3}$	20	20	15	relu
ClinTox	$10^{-3}$	50	40	10	selu	$10^{-3}$	30	70	15	relu
MUV	$10^{-2}$	30	20	3	selu	$10^{-3}$	50	70	10	relu
Sider	$10^{-3}$	50	100	5	selu	$10^{-3}$	10	40	15	relu

**Table 2 ijms-25-06583-t002:** Comparison between the best GNN and the best CGNN configuration on every dataset. Metrics are defined by the OGB [30]: Area under ROC curve (AUROC), root mean squared error (RMSE), and average precision (AP). The best method between the GNNs and CGNNs is highlighted in bold.

Dataset	Metric	GNN	CGNN
HIV	AUROC	0.777	0.792
FreeSolv	RMSE	2.925	2.257
Tox-21	AUROC	0.698	0.726
BACE	AUROC	0.756	0.839
BBBP	AUROC	0.688	0.685
ClinTox	AUROC	0.733	0.925
MUV	AP	0.042	0.056
Sider	AUROC	0.598	0.617

**Table 3 ijms-25-06583-t003:** Comparison between our best model configuration on every dataset and the corresponding SotA method. Metrics are defined by the OGB [30]: Area under roc curve (AUROC), root mean squared error (RMSE), and average precision (AP). The metrics of the SotA methods for each dataset were taken from the paper cited in the corresponding row. The best method between the CGNNs models and the SotA methods is highlighted in bold.

Dataset	SotA Method	Metric	SotA	CGNN
HIV	Graphormer + FPs [31]	AUROC	0.8225	0.792
FreeSolv	GIN [30]	RMSE	2.151	2.257
Tox-21	GIN citeOGB	AUROC	0.776	0.726
BACE	GCN [30]	AUROC	0.792	0.839
BBBP	GIN [30]	AUROC	0.697	0.685
ClinTox	GCN [30]	AUROC	0.913	0.925
MUV	GCN [30]	AP	0.109	0.056
Sider	GCN [30]	AUROC	0.598	0.617

**Table 4 ijms-25-06583-t004:** Datasets used in the experimentation. The number of graphs in each dataset, the number of tasks to be performed, and their type are reported for each dataset.

Dataset	Graphs	Tasks	Task Type
HIV	41,127	1	Binary Classification
FreeSolv	642	1	Regression
Tox-21	7831	12	Binary Classification
BACE	1513	1	Binary Classification
BBBP	2039	1	Binary Classification
ClinTox	1477	2	Binary Classification
MUV	93,087	17	Binary Classification
Sider	1427	27	Binary Classification

**Table 5 ijms-25-06583-t005:** Hyperparameters (and their values) used in each grid search: Initial learning rate (ILR), hidden units of state updating network (HS), hidden units of output network (HO), state dimension (SD), and activation function (AF).

Hyperparameter	Values
ILR	$10^{-2},\;10^{-3},\;10^{-4},\;10^{-5}$
HS	10, 20, 30, 50
HO	20, 40, 70, 100
SD	3, 5, 10, 15, 30
AF	relu,tanh,selu

## Data Availability

All the data used in these experiments were downloaded from the OGB online freely available database: https://ogb.stanford.edu/ (accessed on 6 June 2024).

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
