# Peer review of "Composite Graph Neural Networks for Molecular Property Prediction"

_ijms, 2024, doi:10.3390/ijms25126583_

Round 1

Reviewer 1 Report

Comments and Suggestions for Authors

This work addresses the comparison of performances of several graph-based machine learning models applied to solving classification and regression models using datasets related to molecular graphs. The key matter of investigation is the comparison between Graph Neural Networks and Composite Graph Neural Networks. Although the statement of the problem and reference to the technical implementation of the procedures carried out are clear, the description of the procedure and results requires certain clarifications:

  1. It is not clear, whether the predictive capacity of these networks was already explored: how the data were subdivided into the independent training and test sets, and what were the sizes of these two subsets? It is expected that Table 3 reports the best parameter obtained with the training procedure and Table 4 gives the characteristics of predictive capacity. However, it is not clear from the text whether it is really so.
  2. As for Table 4, it is not clear why different metrics are given for different datasets. In principle, the comparison implies that the same procedures should be applied to the same data and judged with the same criteria.
  3. In general, the sections devoted to results and their discussion should be expanded and made more detailed including, e.g., the relations to the specificity of molecular networks considered, etc. These sections should have a form, which gives clear hints for interested readers on how to operate and what to expect when such a reader would like to apply the proposed method to their own problems. 

Author Response

Reviewer #1

This work addresses the comparison of performances of several graph-based machine learning models applied to solving classification and regression models using datasets related to molecular graphs. The key matter of investigation is the comparison between Graph Neural Networks and Composite Graph Neural Networks. Although the statement of the problem and reference to the technical implementation of the procedures carried out are clear, the description of the procedure and results requires certain clarifications:

Dear Reviewer, thank you very much for working on our manuscript and for all the useful comments and questions.

  • It is not clear, whether the predictive capacity of these networks was already explored: how the data were subdivided into the independent training and test sets, and what were the sizes of these two subsets? It is expected that Table 3 reports the best parameter obtained with the training procedure and Table 4 gives the characteristics of predictive capacity. However, it is not clear from the text whether it is really so.

Thank you very much for pointing this out. Indeed, this is an important point to clarify. We have included a sentence for the predictive capacity of GNNs and CGNNs in the introduction: “While the predictive capabilities of GNNs have been widely studied, the predictive capabilities of CGNNs have not been thoroughly analyzed so far. As a consequence, we propose a comparison between the standard and composite version of the original recurrent GNN model”. Concerning the dataset split, we added the following text to the “Datasets” subsection: “As a consequence, the training--validation--test split is the standard one provided by the benchmark, and it is the same throughout all the experimentation. Moreover, the percentages of training set examples, validation set examples, and test set examples are 80%, 10%, and 10% respectively, for every dataset. The validation set is used for an early stopping module, which prevents overfitting, stopping the training procedure when the validation loss starts to increase. The early stopping module has a patience of 10 and it is applied after every training epoch.” Indeed, Table 3 reports the best parameter configurations and Table 4 the metrics obtained with those parameters on every dataset. We have added the following text to the reference to Table 3: “...reporting the hyperparameters of the best models”, and the following text to the reference to Table 4: “...showing the AUROC, RMSE, or AP of the best model on every dataset”.

  • As for Table 4, it is not clear why different metrics are given for different datasets. In principle, the comparison implies that the same procedures should be applied to the same data and judged with the same criteria.

Thank you for pointing this out. Actually, different datasets imply different tasks to be carried out. Classification tasks require classification metrics, like AUROC and AP, while regression tasks require regression metrics, like RMSE. Moreover, different datasets can have different data balancement characteristics, therefore the OGB benchmark establishes also which metrics should be used for model evaluation on every dataset. We followed the OGB specifications, and applied the correct metrics for each dataset. To better explain this, we added the following text to the reference to Table 4: “... The metric to be used depends on the dataset and is defined by the OGB benchmark itself to best represent the model's predictive capacity depending on the type of task and on data unbalance”.

  • In general, the sections devoted to results and their discussion should be expanded and made more detailed including, e.g., the relations to the specificity of molecular networks considered, etc. These sections should have a form, which gives clear hints for interested readers on how to operate and what to expect when such a reader would like to apply the proposed method to their own problems. 

Thank you for your comment. Actually, we did not use molecular networks. The datasets are just composed of sets of molecular graphs. Our predictions are based only on the molecular graph itself, without focusing on the interactions between different molecules, which are not contained in the OGB benchmarks exploited for this study. The focus is on machine learning models rather than on the data used to demonstrate their results. We have expanded the results and their discussion by providing more contextual information.

Reviewer 2 Report

Comments and Suggestions for Authors

This paper compares the performance of Composite Graph Neural Networks (CGNNs) and standard Graph Neural Networks (GNNs) on molecular property prediction tasks using datasets from the Open Graph Benchmark. The key finding is that CGNNs significantly outperform GNNs on nearly all the datasets by leveraging the heterogeneous nature of molecular graphs, mapping atoms of different species to different node types. This demonstrates that CGNNs' ability to specialize state updating functions for each node type enables them to more accurately predict molecular properties and classify molecules by activity compared to standard GNNs.

My major concern comes from the incremental novelty: CGNNs are not a new architecture proposed in this work, but rather an existing variant of GNNs that is adapted for heterogeneous molecular graphs. The paper does not propose significant methodological innovations in terms of CGNN architectures, training strategies, or evaluation metrics, but rather applies existing techniques to a new domain.

Moreover, this work has no comparison with other state-of-the-art methods. The paper compares CGNNs with standard GNNs, but does not include comparisons with other state-of-the-art methods for molecular property prediction, such as domain-specific architectures. 

Limited discussion of computational complexity: The paper does not provide a detailed analysis of the computational complexity of CGNNs compared to GNNs. While CGNNs have separate state updating functions for each node type, which could potentially lead to increased computational costs, this aspect is not thoroughly discussed.

Author Response

Reviewer #2

This paper compares the performance of Composite Graph Neural Networks (CGNNs) and standard Graph Neural Networks (GNNs) on molecular property prediction tasks using datasets from the Open Graph Benchmark. The key finding is that CGNNs significantly outperform GNNs on nearly all the datasets by leveraging the heterogeneous nature of molecular graphs, mapping atoms of different species to different node types. This demonstrates that CGNNs' ability to specialize state updating functions for each node type enables them to more accurately predict molecular properties and classify molecules by activity compared to standard GNNs.

Dear Reviewer, thank you very much for working on our manuscript and for all the valuable comments and questions provided.

My major concern comes from the incremental novelty: CGNNs are not a new architecture proposed in this work, but rather an existing variant of GNNs that is adapted for heterogeneous molecular graphs. The paper does not propose significant methodological innovations in terms of CGNN architectures, training strategies, or evaluation metrics, but rather applies existing techniques to a new domain.

Thank you very much for pointing this out. Actually, even if CGNNs were introduced in previous works, their capabilities have never been fully studied. Previous works just applied the model to some specific tasks without analyzing it theoretically. The comparison with the standard version of GNNs we propose is a first step in building a theoretical understanding of the capabilities of CGNNs on heterogeneous graphs. In particular, in this work, we are trying to figure out how much a CGNN gains over an equivalent GNN model which is not composite, and therefore not adapted to heterogeneous graph domains. In order to better explain this point, we modified the introduction section, including the following statement: “While the predictive capabilities of GNNs have been widely studied, the predictive capabilities of CGNNs have not been thoroughly analyzed so far. As a consequence, we propose a comparison between the standard and composite version of the original recurrent GNN model”.

Moreover, this work has no comparison with other state-of-the-art methods. The paper compares CGNNs with standard GNNs, but does not include comparisons with other state-of-the-art methods for molecular property prediction, such as domain-specific architectures. 

Thank you for this suggestion. It is indeed important to weigh the findings in light of their relevance with respect to the benchmark tasks and the related literature. We added a Table and a paragraph accounting for this, at the end of the “Results” Section. The State of the Art method for each dataset is shown in comparison with CGNNs. Our model outperforms SotA methods on three datasets out of eight. We also added a sentence to the “Discussion” Section to account for this new part of the results.

Limited discussion of computational complexity: The paper does not provide a detailed analysis of the computational complexity of CGNNs compared to GNNs. While CGNNs have separate state updating functions for each node type, which could potentially lead to increased computational costs, this aspect is not thoroughly discussed.

Thank you for this interesting suggestion. Indeed, Composite models come with a computational overhead with respect to the standard version, which is also dependent on the number of node types in the heterogeneous graph dataset under analysis. We added a detailed discussion of the computational complexity (also in comparison to standard GNNs) at the end of the “Experiments” Section. We also added a sentence to the “Results” and “Discussion” sections to account for this.

Reviewer 3 Report

Comments and Suggestions for Authors

This research presents itself as a critical examination of two well-established models in the field: the Graph Neural Network (GNN) and the Convolutional Graph Neural Network (CGNN). However, while the paper delves into a comparative analysis between these models, it falls short in clearly delineating its unique contribution to the existing body of knowledge.

While the numerical experiments provide valuable insights into the relative performance of GNNs and CGNNs, the absence of sensitivity parameter analysis and complexity assessment creates a gap in the paper's methodological rigor. These analyses are crucial for understanding the robustness and scalability of the proposed models across different datasets and problem domains. Without them, the paper's findings may lack the necessary depth and breadth to make a meaningful impact in the field.

To enhance the paper's significance, future iterations could incorporate a more thorough exploration of sensitivity parameters, examining how variations in these parameters affect the models' performance and generalization capabilities. Additionally, conducting a detailed complexity analysis would provide valuable insights into the computational efficiency of GNNs and CGNNs, shedding light on their scalability and practical applicability in real-world scenarios.

By addressing these limitations and providing a more comprehensive analysis, the paper could significantly strengthen its contribution to the field, offering valuable insights into the efficacy and versatility of GNNs and CGNNs across a wide range of applications.  For all these reasons I propose to reject the paper.

Rejecting the paper based on the outlined reasons seems reasonable, as it's crucial for scholarly publications to meet certain standards of methodological rigor and clarity in contribution. Encouraging the authors to resubmit after addressing the identified shortcomings is a constructive approach.  

Author Response

Reviewer #3

This research presents itself as a critical examination of two well-established models in the field: the Graph Neural Network (GNN) and the Convolutional Graph Neural Network (CGNN). However, while the paper delves into a comparative analysis between these models, it falls short in clearly delineating its unique contribution to the existing body of knowledge.

Thank you very much for working on our manuscript and for all the suggestions and comments provided.

Actually CGNNs are “Composite Graph Neural Networks”, not “Convolutional Graph Neural Networks”. The latter are called Graph Convolution Networks (GCN) in the literature. As explained in the Introduction, Composite Graph Neural Networks are GNNs adapted to process heterogeneous graphs, Each node type in the graph is processed by a different state updating MLP.

Indeed, this is an important point to clarify: in order to better underline the unique contribution of the paper, we modified the “Introduction” Section. In particular, we added the following text: “While the predictive capabilities of GNNs have been widely studied, the predictive capabilities of CGNNs have not been thoroughly analyzed so far. As a consequence, we propose a comparison between the standard and composite version of the original recurrent GNN model.”

While the numerical experiments provide valuable insights into the relative performance of GNNs and CGNNs, the absence of sensitivity parameter analysis and complexity assessment creates a gap in the paper's methodological rigor. These analyses are crucial for understanding the robustness and scalability of the proposed models across different datasets and problem domains. Without them, the paper's findings may lack the necessary depth and breadth to make a meaningful impact in the field. To enhance the paper's significance, future iterations could incorporate a more thorough exploration of sensitivity parameters, examining how variations in these parameters affect the models' performance and generalization capabilities. Additionally, conducting a detailed complexity analysis would provide valuable insights into the computational efficiency of GNNs and CGNNs, shedding light on their scalability and practical applicability in real-world scenarios.

Thank you for your comment. Of course, the model robustness and scalability are important aspects to take into account. We added a detailed discussion on the computational complexity of the models, also in relation to the experiments we carried out, at the end of the “Experiments” Section. Concerning the sensitivity, this metric was not taken into account because it is not provided by OGB evaluators and it is never used on OGB benchmarks. The OGB defines the best metrics to be used for model evaluation on each benchmark dataset, according to the characteristics of the data and of the task(s). We strictly followed these metrics, also because they are used by the relevant literature on OGB benchmarks, but mainly because these metrics provide the best evaluation method for the model performance and its robustness to data imbalancement and other possible biases. The main metrics are indeed AUROC for classification tasks and RMSE for regression tasks. Some particular multi-class multi-label tasks, such as the MUV dataset, instead require Average Precision as the AUROC evaluation would always give very low scores for every model. We added a summary of this to the paper in order to better explain the metrics choice.

By addressing these limitations and providing a more comprehensive analysis, the paper could significantly strengthen its contribution to the field, offering valuable insights into the efficacy and versatility of GNNs and CGNNs across a wide range of applications.  For all these reasons I propose to reject the paper. Rejecting the paper based on the outlined reasons seems reasonable, as it's crucial for scholarly publications to meet certain standards of methodological rigor and clarity in contribution. Encouraging the authors to resubmit after addressing the identified shortcomings is a constructive approach. 

Thank you very much for working on our manuscript. We believe that the paper has greatly improved, also thanks to your constructive comments and suggestions.

Round 2

Reviewer 1 Report

Comments and Suggestions for Authors

The authors took into account all questions and properly revised the maunuscript made its approaches and results clear for readers. It can be accepted now. 

Minor editorial issue, which must be corrected:

in the newly added part (lines 337-341), there are uncompiled LaTeX labels ("?" signs instead of numbers of references to a table and literature sources). 

Reviewer 2 Report

Comments and Suggestions for Authors

Thanks for the reply, my concerns have all be addressed.